# A SWOT: Thematic Analysis of Pedagogical Practices at Inclusive School of Pakistan

Mahwish Kamran [1], Nazia Bano [2] and Sohni Siddiqui [3,*]

1 Department of Education and Social Sciences, Iqra University, Karachi 75500, Pakistan;
mahwish.siddiqui@iqra.edu.pk
2 Independent Researcher, London, ON N6A 3K7, Canada; nazia.bano@iuk.edu.pk
3 Department of Educational Psychology, Technische Universität Berlin, 10587 Berlin, Germany
* Correspondence: s.zahid@campus.tu-berlin.de

**Abstract:** In Pakistan, where the subject of special/inclusive education is still taboo, there is a need to promote inclusivity in education. However, the journey begins at the grassroots level by accommodating children with disabilities in a mainstream setup at the primary level. This paper presents the findings of an exploratory research study conducted in an inclusive private primary school in Karachi, Pakistan. This case study research draws on the pedagogical practices of classroom teachers in a private primary inclusive school in Karachi where children with disabilities study alongside their peers who do not have special educational needs or disabilities. The research study aimed to explore the strengths, weaknesses, opportunities, and threat factors that could optimize the teaching and learning process of children with special educational needs (CWSN) or children with disabilities (CWD) in the context of an inclusive school located in Karachi, Pakistan. Through an analysis of 16 semi-structured interviews and multiple classroom and field observations, teachers' understandings of their school's institutional values and their pedagogical practices to accommodate children with disabilities and inclusion were explored. The interviews were transcribed and analyzed using a SWOT–thematic qualitative method. The results of the SWOT analysis indicate how an inclusive school caters to the strengths of CWD and provides them with opportunities to sustain themselves in an educational setup.

**Keywords:** inclusive school; children with disabilities; pedagogical practices; SWOT

## 1. Introduction

The notion of inclusive education is not limited to children with disabilities only; it applies to all children, regardless of their gender, culture, personality traits, or economic status of their parents [1]. An inclusive school accommodates children with disability in the same classroom alongside their peers who are without a disability [2]. Accomplishing an inclusive school community requires establishing an inclusive school culture and environment and the participation and cooperation of all stakeholders within the school community. "It implies a process of enculturation of learning and teaching whereby educators and communities have to shift from a set of fixed norms, beliefs, customs, and practices that encourage the preservation of the status quo to one that encourages transformation, including building a commitment to change and providing the support that promotes and maintains change" [3]. According to Hyde et al. (2014), inclusion is not something that "just happens", but rather something that entails careful thought and groundwork, executed with proper attitudes, support, and adaptations in place [4]. This is also echoed by Ewing et al. (2018), who stated that effective inclusion can only be a reality when the interaction of certain key factors takes place [5]. One of the significant factors mentioned is that appropriate support for teachers is imperative to the success of inclusive education. Ewing et al. (2018) also found in the study that teachers needed administrator support, "both in terms of resources and emotional support—to feel comfortable with differentiating curriculum, pedagogy, and

assessment" (p. 99) [5]. Considering this statement, it is important to understand that the role of schools is pertinent. It covers all areas of the school including pedagogical practices, leadership roles, student–teacher interaction, and peer-to-peer interaction. It also includes the provision of material resources and how all these aspects have facilitated the school's progress toward setting up an inclusive school [6].

The word 'pedagogy' is frequently used in teaching and learning. Alexander, as cited in [7], refers to the act of teaching together with its attendant discourse. It is what one needs to know, and the skills one needs to command. The pedagogical practices adopted in diverse classrooms can play a pivotal role in carrying out successful inclusion [8]. Therefore, the current case study explores the pedagogical practices of teachers accommodating children with and without disabilities in an inclusive school. Inclusive education means that all children of the same age group must be educated in the same classroom, regardless of their physical, intellectual, and emotional abilities [9,10]. Scholars have discussed whether teaching children with special educational needs entails particular pedagogical practices or whether teachers can use regular teaching methods. Garrote et al. (2020) stated that all students have needs that are mutual to all and needs that are exclusive to them as individuals [11]. However, in any case, it is a child-centered inclusive pedagogy that effectively responds to the needs of all students [8]. The same pedagogy was witnessed in the Praxis School.

In this regard, few schools in Pakistan have taken the initiative to include children with disabilities by adopting an inclusive pedagogy and accommodating them with children without disabilities in a regular classroom setup [12]. Moreover, in Pakistan, inclusive education is still evolving and institutions are not ready to cater to students with disabilities [13]. However, Shaukat (2022) pointed out that child-centered inclusive pedagogy was missing, and this is the reason for the exclusion of children with disabilities from the mainstream setup [14]. The Praxis School, which was the research site, ensures the practice of child-centered inclusive pedagogy. It is a private primary school with grades I to V accommodating both children with and without disabilities. In this case study, a SWOT analysis was carried out to explore the strengths, weaknesses, opportunities, and threats related to this Praxis School. This study is intended to present significant findings that will help to build a tangible foundation for revamping current inclusive educational practices, as well as to direct future educational policies to create inclusive schools that meet the needs and requirements of children with disabilities.

## 2. Literature Review

### 2.1. Theoretical Background: Ecological Systems Theory

Bronfenbrenner's ecological systems theory (1977) is based on the concept that the environment of a child is basically a sequential arrangement of structures, and this structure influences children's development [15]. All levels in this structure are interconnected hierarchical systems where the impact of one system on development depends on its interaction with the others. The study is guided by Bronfenbrenner's (1977) ecological system theory and factors that affect the application of inclusive practices [15]. There are four levels: macrosystem, exosystem, mesosystem, and microsystem. This framework clearly shows the relationship among the four layers that are important for inclusion. This is also a guide for educators that makes them aware of what must be considered during the application of inclusive practices. The theory clearly indicates that inclusion can be perceived from an ecosystemic point of view [16]. Bronfenbrenner (1977) also highlighted the basis of a child's development, as it is in the hands of the interaction that takes place between the child and its social milieu [15]. This interaction is a prerequisite of inclusion that can be a contributing strategy for supporting children with disabilities [17]. To determine how to proceed, schools play a crucial role in early diagnosis, as they need to locate the problem as early as possible.

Bronfenbrenner (1977) believed that school plays a significant role, as education roots are in schools [15]. From the point of view of the five ecological systems as well as from

the point of view of a connection between psychological and educational theory and early educational curriculums and practice, the focus of the theory is on the child's development, and all that takes place within and among the five ecological systems is done to benefit the child in the classroom [18]. Another aspect is that a school endeavor, both inside and outside of the school, is a society and education is a social process. To consolidate the development of ecological systems in educational practice, according to the theory, all stakeholders, including leaders, teachers, parents, and students, should maintain good communication with each other and work in collaboration to support the child [17]. Teachers should also try to understand the situations families of students may be experiencing. This includes social and economic factors that are part of the various systems. According to ecological system theory, if stakeholders including parents and teachers have a positive relationship, this can shape the child's personality in a productive way [14]. Similarly, the child becomes an active learner engaged both academically and socially. They work as a team with their peers and become engrossed in meaningful learning experiences to enable holistic development. The pedagogy adopted in schools needs to be such that it can pave the way toward emancipation. On the basis of the ideas of Bronfenbrenner (1977), it is believed that a quality school is one that assures the student can learn; does not give up on placing him/her in the human circle; takes rights of instructional practices as a prospect of the difficulty of uneven processes of contribution in life in society; undertakes the assurance of education for all learners, irrespective of their financial, personal, social, or emotional conditions; nurtures a moral obligation to allow ways for learners to make their cultural production; and takes multiplicity as a rich prospect of human progress and not as a component that hinders education [16]. In this regard, it is imperative to explore the school that is working along the lines of allowing ways for learners to make their cultural productions. The study is important as it is based on the notion of exploring an inclusive environment in which all learners have the same opportunity to be successful irrespective of ethnicity, race, belief, and special needs. It is also important to understand that Kelly and Coughlan (2019) used constructivist grounded theory analysis to develop a theoretical framework for youth mental health recovery and found that there were many connections to Bronfenbrenner's ecological systems theory in their own more recent theory [19]. Their theory suggested that the elements of mental health recovery are embedded in the 'ecological context of influential relationships' which fits with Bronfenbrenner's theory that the ecological systems of the young person, such as peers, family, and school, all help mental health development [19]. It is the same in the case of the current research study, in which the theory is linked to the positive intellectual and social health development of children with special educational needs.

### 2.2. Challenges Associated with Inclusive Education Setups

Educators globally have acknowledged the negative consequences of segregating children with disabilities from regular classroom settings or mainstreaming [20]. Nevertheless, incorporating children with special needs remains challenging because of various reasons, including a shortage of teachers with special education competencies. It is pivotal to understand that a deficit exists in the number of teachers possessing the essential competencies in inclusive education. This scarcity carries implications for the education of children with disabilities, indicating that inclusive schools may be better equipped to address the needs of these students [21]. Teachers' ability to teach students with special needs in regular classrooms is mainly affected by their qualifications and teaching methods. This emphasizes the importance of more training and improved teamwork between special education and regular education [20].

For the effective integration of children with special education needs, particularly those with intellectual and developmental disabilities, it is crucial to implement practices that have been validated through rigorous research [20]. Some research studies argue that the international orthodoxy of inclusive education might not be directly applicable to the educational systems of low-income countries. These countries might find it challenging

to transition from almost no educational provision for disabilities to a fully integrated provision [20]. Inclusive schools continue to play a vital role in the education of both children with and without disabilities. One significant benefit is that children without disabilities cultivate a humanistic approach when studying alongside their peers with disabilities. Additionally, they acquire qualities of patience and compassion [22]. Despite these advantages, some schools hesitate to embrace inclusive education due to the challenges outlined above [23]. The literature on inclusive education highlights numerous benefits for both groups of students. However, unlocking these advantages is contingent on schools effectively addressing challenges, such as a shortage of trained teachers, implementing inclusive teaching practices, and securing adequate resources [24].

### 2.3. A Move toward Inclusive Education in Pakistan

The Government of Pakistan is a signatory, endorsing international documents, for instance, the Salamanca Statement [25]. The National Policies of 1986 and 1988 for People with Disabilities had already presented the ideas of mainstreaming and integration. However, the Government of Pakistan has not always been able to execute these policies at the grassroots level. Consequently, the philosophy of inclusive education found its way into the national policy on disabilities [26] and into the National Plan of Action, launched in 2006, with the purpose of ensuring its enactment at a future stage [2].

Focusing on the Salamanca Statement [27], the wish of the Government of Pakistan has been to embrace the basic norms of an inclusive school in which all children learn together, irrespective of their abilities or differences. However, it has been accepted by educationists and researchers that inclusion is a multifaceted and ever-progressing practice, demanding constant restructuring in guiding principles and preparation [6]. Its attainment depends on the staffing and preparation of teaching staff, the provision of support services, and community involvement, as well as the allocation of sufficient resources [27].

To these ends, the Government of Pakistan, in its former policies and documents, made pledges (see Figure 1) to support special and inclusive education [2,28]. The Provincial Education Department in Sindh, Pakistan, presented a project of founding small integrated units in mainstream schools, but this was also unsuccessful due to the insufficient training of teachers. This indicates the issue of sustainability, as the programs were initiated but were not successful.

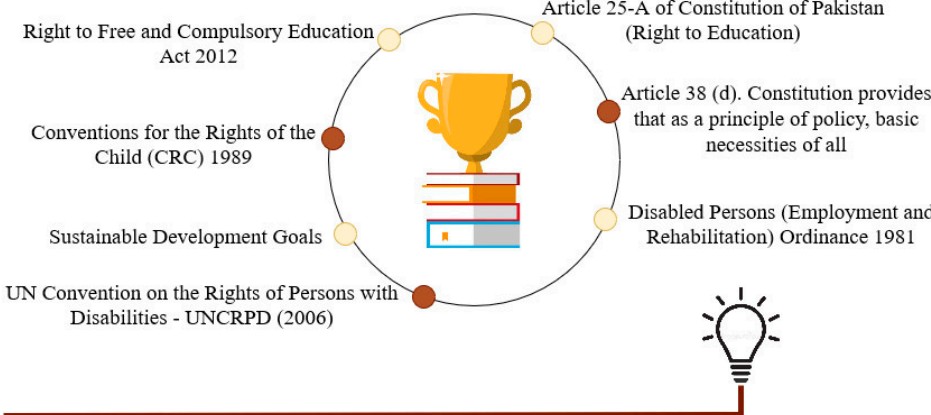

**Figure 1.** National and International Commitments for Special and Inclusive Education Pakistan (developed by the author). Articles quoted in the diagram are available in National Education Policy 2017 [1].

Similar is the case in which many schools that initiated promoting inclusive educational practices were not successful. The examples can be witnessed in Karachi, where an inclusive setup stopped providing its services [12]. Pakistan has signed and ratified the UN Convention on Human Rights and the Rights of People with Disabilities (UNCRPD), which calls for equal rights and opportunities for all people including those with disabilities [29]. It became the responsibility, after the 18th Constitutional Amendment, of both federal and provincial governments to commence immense and exhaustive planning in order to accomplish the provision of inclusive education. However, the schools that are excelling in the field of inclusive education need to be explored, as they remain largely unexplored [30].

### 2.4. SWOT (Strengths, Weaknesses, Opportunities, and Threats)

SWOT (strengths, weaknesses, opportunities, and threats) is a strategic approach that emphasizes strengths and strives to understand an organization and its working environment by including the voices of relevant stakeholders [31]. It is frequently used to analyze the strengths, weaknesses, opportunities, and threats related to educational programs implemented in different educational institutions. Although the SWOT analysis was originally employed in the field of management studies, its applications extend beyond specific domains and have been applied in educational settings [32,33]. This also includes studies that assess the strengths, weaknesses, opportunities, and threats of inclusive schools [34].

The research study created a SWOT framework based on the strengths that exist in a Praxis School, which is a private school (K–V), offering elementary, senior, and inclusive education that has enabled them to sustain themselves as an inclusive school for more than 25 years. The school's uniqueness lies in the fact that it is the only inclusive school that has the ability to provide education to members of the disadvantaged socio-economic class, as well as maintain its success over time. Further details about the school are added in a research context in the next section.

In order to ensure the successful inclusion of CWD, opportunities need to be focused on by removing barriers and offering the least restrictive environment. These opportunities are a sign of hope that enables them to achieve long-term goals and maintain success. As a result, the Praxis School can overcome coercions and pave the way toward inclusion (Figure 2).

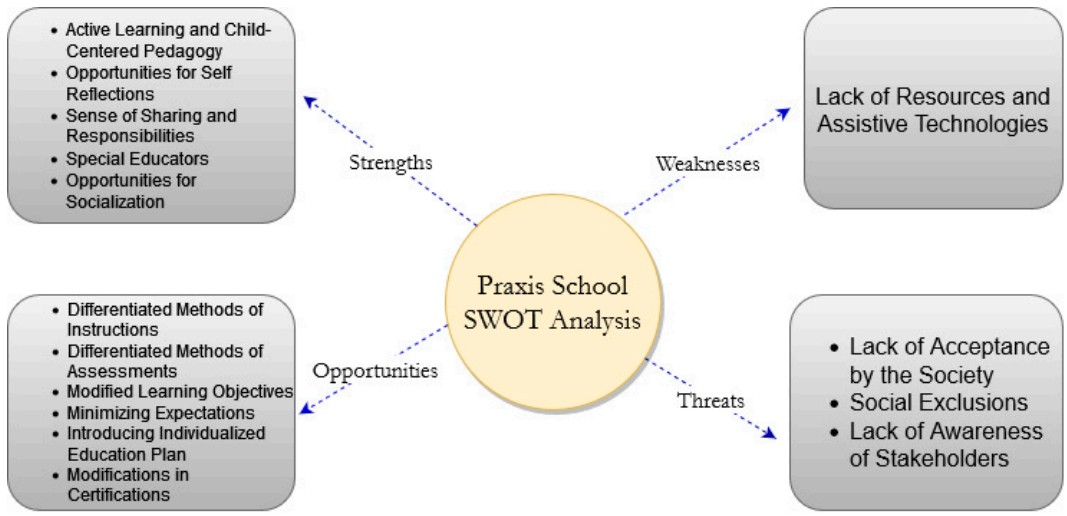

**Figure 2.** SWOT analysis model (developed by the author).

## 3. Research Methodology

### 3.1. Research Design

The research employed a qualitative case study design, leveraging the benefits of this approach to acquire a profound comprehension of a specific event or individual through direct observation and data collection in a natural setting [35]. This case study is confined

to a singular location or a "bounded system" [36]. According to Yin (2013), a single case study design should encompass distinctive circumstances [37]. In this particular research case study, the sole location under examination is an inclusive school that stands out as an extraordinary case, maintaining an inclusive culture for a remarkable 25-year period. The authors refer to that school as the Praxis School (pseudonym) throughout the text.

### 3.2. Sampling

In pursuit of the objectives of the current case study research and to address the research inquiries, the researcher employed a purposive sampling technique. This method involves selecting a sample based on information about the population and the specific parameters of the research study [38]. In the context of a case study, purposive sampling entails choosing subjects based on specific characteristics [39]. Consequently, participants were chosen by establishing selection criteria, which are detailed in Table 1 for all stakeholders, encompassing teachers, administrators, and coordinators.

**Table 1.** Research instrument—interview and focal group discussion guidelines.

| Participants in Interviews | Selection Criteria | Themes of Interview Questions | |
| --- | --- | --- | --- |
| | | General Themes | Specific Themes |
| Administrator (N = 1) | • Role in establishing the school<br>• Knows the historical context | | • To understand the historical context<br>• To gain insight into conceptualizing the idea<br>• To collect information in terms of collaborating with all stakeholders<br>• To explore factors that enabled them to sustain |
| Coordinators (N = 3) | • Experience of 10 years or more<br>• Must be catering to different types of disabilities | • Content<br>• Practices/approaches<br>• Coping mechanism<br>• transition and sustainability<br>• Struggles/limitations/endurance | • School support<br>• Teaching strategies<br>• Teachers' challenges and coping strategies<br>• Their understanding and teaching of special needs assessments<br>• Lesson planning |
| Teachers (N = 12) | • Teaching experience of 10 years or more<br>• Grades they are teaching (grades I to V)<br>• Presence of special needs students in their class<br>• Professional development courses undertaken | | • Nature of the interactions<br>• Support in terms of pupils' behavioral, social, and emotional needs/development<br>• Support provided to teachers |

### 3.3. Research Context

The current research is situated within the private education system (K–V), established 25 years ago with a mission to eliminate both apparent and hidden barriers for children with mild to moderate disabilities, allowing them to partake in a school experience alongside their regular peers. What sets this school apart is its unique status as the sole inclusive institution capable of delivering education to the economically disadvantaged. This is while maintaining enduring success. Commencing its journey in 1996, the school offers elementary, senior, and inclusive education, employing English as the primary medium of instruction and including Urdu as a subject. Situated in a densely populated city area, the purpose-built school features a teaching and student body primarily hailing from middle-

to lower-middle-class backgrounds, indicative of the economically challenged households of the students' parents. The teaching staff, predominantly female, includes a few male teachers specializing in subjects such as sports and taekwondo. The school's infrastructure encompasses spacious, well-lit classrooms with ample cross ventilation. Class sizes vary, with grades 1–5 organized into sections, each covering subjects like English, math, science, social studies, Urdu, Islamiat, arts, computers, and sports. Resources range from textbooks and guidebooks to whiteboards, with additional facilities like a library, workspaces, a computer lab, and a science practical lab. The school accommodates various forms of disabilities, with a particular focus on integrating children with mild disabilities, as outlined in Table 2. Notably, the school assesses adaptive behavior using the adaptive behavior diagnostic scale, identifying behavioral deficits in children with intellectual disabilities, autism, learning disabilities, and more. The presence of a psychologist aids in diagnosing specific disabilities, enabling the creation of tailored instructional plans for children with special educational needs.

**Table 2.** Statistics on the school.

| Students without Disabilities * | Students with Disabilities in the Integrated System | | Students with Disabilities in the Inclusive System | | Total No. of Students Registered |
|---|---|---|---|---|---|
| | 40 | | 74 | | |
| 346 | Male = 28 | Female = 12 | Male = 62 | Female = 12 | 460 |

* Autism, Down's syndrome, ADHD, global developmental delay, hearing impairment, intellectual disability, learning disability, developmental delay, and cerebral palsy.

*3.4. Ethical Considerations*

Current research views potential participants as collaborators in a joint venture, built upon a foundation of mutual respect and trust [40]. It is crucial to highlight that an ethical researcher consistently values and respects the dignity of respondents, thereby minimizing harm and upholding ethical conduct throughout all stages of the research, including the writing-up phase and beyond [40]. This relationship is grounded in informed consent, which was diligently obtained to affirm the absence of harm or threat to participants. Efforts were made to establish trustworthiness and confidence between the researcher and participants, with a particular emphasis on ensuring the confidentiality, privacy, and anonymity of the participants through a formal letter. All transcripts were securely stored, and participants who signed the consent form were verbally assured that their identities would not be disclosed to anyone. Importantly, before initiating the research study, an ethics review form was completed to sensitize and address concerns related to respondents' anonymity. This form, developed by the university's ethics review board, covered areas such as informed consent, confidentiality, and data storage. Consequently, participants signed informed consent forms and received a detailed letter of information outlining the research study's objectives and significance. It is important to mention that to ensure confidentiality pseudonyms are used throughout the manuscript.

*3.5. Data Collection Tool*

The primary method employed for data collection involves observations with field notes. According to Patton (1999) and Denzin and Lincoln (2005), observation can be a crucial and highly effective method during a research study, with field notes serving as a valuable means of recording these observations [39,41]. However, the principal research tool utilized was one-to-one semi-structured interviews with all stakeholders. These included professionally developed teachers with more than 10 years of experience (N = 12), an administrator who established the setup and was familiar with the historical context of the school (N = 1), and coordinators with more than 10 years of experience to tackle different types of disabilities (N = 3). In qualitative research, semi-structured interviews stand as a widely adopted data collection method [42], offering the opportunity to gain insight and gather in-depth information. Guidelines for these semi-structured interviews were

developed before their initiation, aligning with the themes identified in the literature. These guidelines encompassed questions pertaining to teaching strategies, lesson planning, and other relevant aspects (refer to Table 1).

## 4. Data Analysis

To establish familiarity with each text, the transcripts and field notes underwent multiple rounds of reading and rereading. Data analysis employed open coding, following the approach outlined by Strauss and Corbin (1998) [43]. To discern patterns and clusters within the dataset, manual coding using highlighter pens was employed, constituting the process known as 'data reduction', the initial phase of data analysis [38]. Subsequently, the data were organized into systematic chunks, referred to as 'data display', marking the second stage in the analytical process [38]. According to Miles and Huberman (1994), the third stage involved 'conclusion drawing/verification', wherein themes are analyzed, aligning with the process outlined by Boyatzis (1998) [38,44].

The same systematic process was applied to data analysis to ensure the consistency and reliability of findings. Triangulation, peer debriefing, and member checking were implemented to safeguard the social validity of the findings with participating stakeholders, as advocated by Lincoln et al. (1985) [45]. Themes and subthemes were categorized based on the SWOT analysis model, identifying strengths, weaknesses, opportunities, and threats (see Figure 3).

**Open Coding**
Coding entire datasets

**Axial Coding**
Aggregating similar codes together to form potential categories

**Selective Coding**
Identifying core categories and themes

**Figure 3.** Strauss and Corbin's (1998) method of data analysis [43].

## 5. Research Findings

### 5.1. Pedagogical Practices as Strengths

The findings suggest that teachers at the Praxis School believed in imparting active learning in inclusive classrooms. This is the strength of the school, as they train teachers to carry out active learning. The teachers believe that the outcomes of active learning were positive in terms of the academic, physical, social, and emotional development of children with disabilities. It involves four interdependent components, as illustrated in Figure 4.

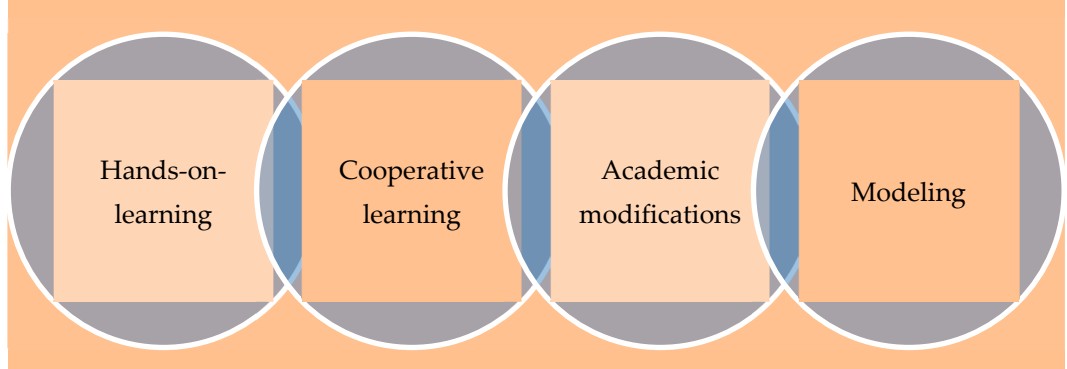

**Figure 4.** Interdependent components of active learning (developed by author).

During the classroom observation at the Praxis School, it was noticed that a teacher was teaching fruits to children and taught them using multiple ways, for instance, through flashcards, models, drawing a tree, and asking them to place fruits on a whiteboard. All these activities were mentioned in their lesson plan as well. However, the children with disabilities still needed more practice. Then, the teacher decided to invite a fruit seller and discussed this with management. The next day it was observed that the fruit seller came and the children bought fruits. The children took them in their hands, understood it well, and then came back to the class and quickly performed all activities related to fruits. One ADHD student stated, "*All fruits have different colors and shapes*" (Saif). Children with disabilities showed an exceptional performance, and the strategy worked well. Children with special educational needs and those without special educational needs both learned through hands-on learning and the experience of holding the fruits in their hands. During the observations, it was witnessed that they reflected on it as well. A teacher stated, "*Learning through reflection on doing*" (Kainat). Here, it has taken the form of experiential learning.

Shaukat (2022) stated that active learning cannot take place without cooperative learning. Learners share responsibilities and resources in working toward common goals [14]. A teacher in grade IV stated:

*When it is well-implemented, just like the group work carried out by us, it allows a teacher to spend more time with individuals and small groups. Alternative teaching in school presumes that one teacher works with a small group of a class, while the helper teacher (who is trained) works with the remaining students. This ensures group work is effective in inclusive classes, particularly when teaching children with disabilities. It is best to avoid making groups based on ability levels (Aiman).*

The abovementioned example was observed during a classroom observation, which is a perfect example of cooperative learning. In the above statement, it seems group work is modified as alternative teaching is carried out. It was observed that a modified or alternative version of group work was used with a small group of students who are children with special needs (CWSN) or need more attention, and the other teacher dealt with the entire class. The smaller group of CWSN was busy matching vocabulary words with pictures and the larger group was busy constructing sentences from the same vocabulary words. It was also observed that the CWSN group constructed sentences and the other group performed matching. However, a group from the CWSN made simple sentences with the help of a teacher, and the children without special needs made complex sentences. This was done just to ensure inclusivity and equity. Inclusivity and equity go hand in hand.

Group work needs to be carried out carefully to ensure the activity is purposeful for all, and teachers need to be available during the tasks to provide support where needed [46]. Each child can perform a role. It was observed during a group work activity that the teacher taught parts of the body to children. She drew a circle on the board and provided the same worksheet to all children (the worksheet was attached to their lesson plans). She drew

different facial features. Then, on the chart paper, she drew a big circle. The children sat in a circle around her on the floor. They had paper and a pencil. Everyone drew different features of a face and stuck them on the chart. In this group work, not only did socialization improve, but students mingled, rapport was developed, and every student contributed.

Anna and Angharad (2021) stated that for many students with disabilities and many without, the key to success in the classroom lies in having appropriate adaptations, accommodations, and modifications made to instruction and other classroom activities [46]. At the Praxis School, modification was witnessed. A teacher's statement highlights modification for teaching CWSN:

> *For instance, if a regular child learns the grass, root, and sky letters. These children simply learn letters, as the concept of grass, root, and sky seems unnecessary to them. The teacher taught them by saying to begin with a red line in a copy [In English copies there are lines red on the top for beginning with a capital letter]. This [sky, grass, and root letter] is an extra burden on CWSN, and it seems unimportant as well so we skipped it from the syllabus and this is a modification (Amber).*

An English teacher modified the teaching strategy, as she believed unnecessary concepts were useless. In the abovementioned example, the teacher did not tell a child with a disability about grass, root, and sky letters. In English, students are taught these just to understand the concept of capitalization. The teacher simplified it. She taught them letters by asking them to begin with a red line if it is a capital letter or a blue line for a small letter. She put dots on the lines to show them where to begin and where to end. Later they did it on their own, as they understood the concept of using different lines for writing letters.

A very different perspective on modification was observed. In the case of one child, a different sort of modification was required and which was not a strategy related to modification but included activities to improve their fine motor skills. A teacher stated:

> *Teaching in an inclusive setup or modification is not rocket science. It is all about understanding a child. There was a child, Ray, who was good at English. I observed that he could say "a" but could not write it despite holding his hand. I told his mother that he had a problem. He needs muscular exercises (Amber).*

In the abovementioned case, the child did not need modification in the first place; however, they needed muscular exercises. It is very important to understand that modification can be supported by muscular exercises. For instance, at the research site, a modified activity with playdough was performed well by a child, as they had muscular exercises. In the activity, they needed to mold the dough into shapes. The children without a disability had to make five shapes. The CWSN were instructed to make two as part of the modification. As his holding capacity was worked on through muscular exercises, he, therefore, performed the activity well. These research findings resonate with universal findings [46], since in the view of teachers at the school there are no specific teaching strategies/pedagogical practices for children with disabilities. Inclusive teachers use and adopt a child-centered pedagogy to meet the needs of all children.

Sharma et al. (2013) stated that imitating others' behaviors is called modeling. Modeling is a very powerful behavior management strategy observed at the Praxis School [47]. By demonstrating social skills, such as saying 'please' and 'thank you', and helping others, teachers can, thus, affect the behaviors of children. The teachers modeled it and frequently used 'magic words', for instance, 'please', 'thank you', 'sorry', and 'excuse me'. All of these were used by all of them and observed during school visits. It was a perfect example of training and preparing their children to be, as a teacher stated, "*Well-mannered members of society*" (Kainat).

One of the most effective ways, as shared by teachers and as observed, was the reduction in inappropriate behavior in the classroom and promotion of good behavior by praising them when they exhibited positive behaviors. This proved to be the most effective, and the teacher modeled it. The teacher provided them with incentives by sending them to physical exercise or music classes. This encouraged the children to stick to positive

and acceptable behaviors in class. A teacher said, "*Here it is needed that teachers praise and encourage good behaviors continuously*" (*Farina*) and this worked miraculously well. The teachers used specific verbal praise which proved to be extremely effective: "Sadiq, you are such a disciplined child" or "Amin, you are working peacefully!". This sort of positive reinforcement was witnessed at the Praxis School. A science teacher shared:

> *I always ask parents to begin with small reinforcements. I never encourage parents to take them for outings daily. For instance, a fun land or a play land. Let's just give them a star, a card, a sticker, or a candy or prepare their favorite food. What I do is I tell them a story, and send them to a TV room, for PE. Let them play with toys. These are all different kinds of reinforcement (Yasra).*

In the above example, the teacher emphasized small reinforcement so that students do not demand much; otherwise, they might become stubborn, have tantrums, or aggravate some other behavioral issues. However, behavior charts were hung on the walls to manage and reward good behavior. A teacher stated, "*These behavior charts keep them disciplined and motivated*" (Rushna). Therefore, teachers train parents on how they should deal with them and model appropriate behavior.

*5.2. Research Findings–Weaknesses*

The Praxis School faces challenges, and they become weak areas, as they hamper productivity. Among these areas, a lack of resources is a major concern. It is important to mention that the terminology 'assistive technology' means a device or a tool, any equipment, or even a software program that can expand the functionality of persons with disabilities (PWD), children with special educational needs (CWSN), or children with disabilities (CWD). It could also include tools that help them accomplish their routine tasks self-reliantly. It can consist of mobility devices, for example, wheelchairs, and ground-breaking software programs that can support diverse disabilities, such as learning disorders and audio–visual impairments, to name a few [48].

A very important finding was noted during a classroom observation regarding a special chair designed for a child living with autism, who cannot balance himself and is in danger of falling using a regular chair. These chairs are assistive technology (AT) devices that can be used for facilitating CWSN. However, the school has a limited budget and a lack of funding; therefore, they do not have an ample number of such chairs.

The findings of the current research study helps in gaining awareness of the usefulness of assistive technology devices in catering to children with special educational needs. The current research study explored how assistive technology can play a role in not just decreasing the academic dependence of children with special educational needs but also helping them to achieve social interdependence. This idea resonates with the literature, as Anna and Angharad (2021) emphasize that the incorporation of assistive technology can make children with disabilities independent [46]. A coordinator of an inclusive setup stated:

> *Assistive technology has many benefits and its incorporation can be multifunctional. Children with special educational needs can significantly benefit from assistive technology. With the help of assistive technology, children with disabilities can overcome difficulties in all four basic skills: speaking, reading, listening, and writing, also, mathematical reasoning and problem-solving (Zareen).*

Similarly, in accordance with the No Child Left Behind (NCLB) Act, the US Department of Education is accentuating technology incorporation to enrich the effectiveness of instructional intervention and, in turn, academic success. Academic success is a major benefit because when the use of software and devices solve children's problems, they improve academically [49]. The coordinator of the integrated setup stated:

> *In order to implement assistive technology in the school for catering to special needs children, the school must have a budget. Teachers who cater to children with special needs are given augmented tasks in inclusive education classrooms. The struggle is therefore*

*planned to connect children with disabilities to classroom activities that their peers are relishing, resulting in a sense of achievement, collective actions with distinct outcomes, and unbiased didactic knowledge. However, it is not possible without assistive technology. Additionally, the use of assistive technology can also lessen frustration, increase zeal, foster a feeling of peer acceptance, and develop efficiency in school and at home. It is therefore required to emphasize the fact that funding is needed for the application of assistive technology.*

A variety of assistive technology devices and tools are available that, with cautious preparation planning and management, can help students with special educational needs. For instance, hearing aids can support children with a hearing impairment [46]. The findings suggest that technological interventions in AT devices support all types of disabilities, including hearing and visual impairments, autism spectrum disorder, and physical disabilities. Children with special needs can lead an independent life not just because of their academic independence but also because of their social independence. If they are active users, they can play a role in the economic development of a country. Through assistive technology, we can transform education for children with disabilities so that they can play an active role in society as valued human resources [10]. However, the dilemma is that schools lack AT because of financial constraints. They need funds and the support of the government.

*5.3. Research Findings–Opportunities*

Being proponents of inclusion, the coordinator of the inclusive setup introduced differentiated methods of assessment into their instruction to support CWD for the purposes of achieving academic success and not being left behind by their peers. These methods are illustrated below in Figure 5. She explained that two key methods of assessment are academic success, which is based on the attainment of modified learning objectives, and acquiring skills and competencies. Another is differentiated modes of assessment. It is important to note that as learning objectives are modified, therefore, the methods of assessment are also modified. The assessment is designed based on the modified content taught to them, and these are discussed in detail below.

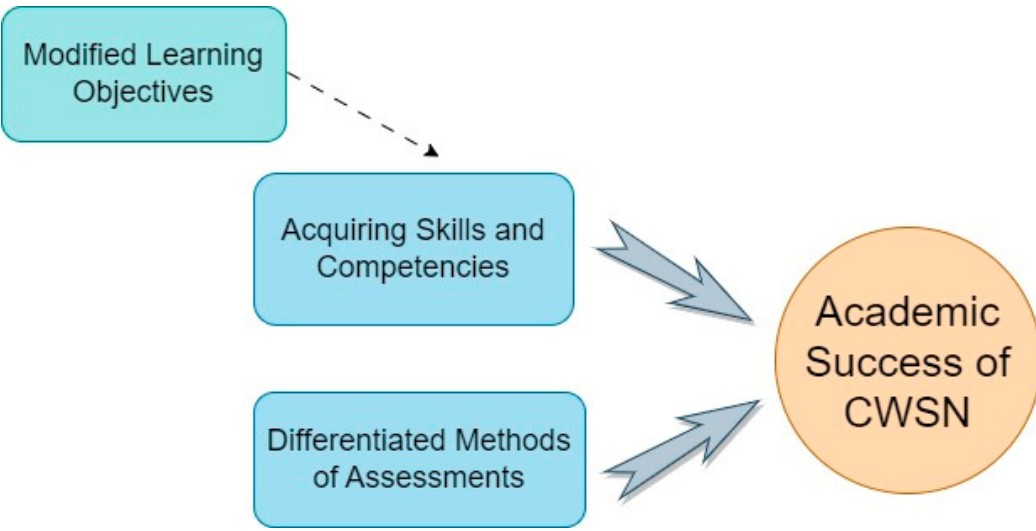

**Figure 5.** Academic success of CWSN (developed by the author).

To explain the differentiated methods of assessment, the coordinator of the inclusive setup reported the example of 'the verbal assessment' method, stating the case of a CWSN. According to her statement:

*A child completed grade II at another school before coming to us. After observing him I felt that the previous school kept him verbal and the mother was insisting that he should start with written tasks. However, the child was comfortable with verbal communication as he had some shivering issues. As soon as a written task is assigned because of pressure he starts shivering. The task that a regular child does in 1 h he does in two/two and a half hours. His IQ was good, except for mathematics. He also has some issues walking. His body shivers under pressure. If such issues are there then we prefer verbal assessment, just to find out where he stands and what we need to work on (Zareen).*

By analyzing this finding, it can be concluded that alternative methods of assessment are beneficial to this child, as in this case verbal was preferred over written evidence. As such, explaining the importance of differentiated methods of assessment, one teacher said:

*Alternative modes of assessment are one of the reasons for securing good grades as they enable them to cater to the individual needs of learners, and when these needs are understood they can be successfully addressed. This method enhances the learning outcomes of children with special educational needs (Isra).*

The coordinator and a teacher of an inclusive setup also explained the entire process of a formative assessment. The coordinator stated:

*We have mid- and final-term papers. We have monthly tests during the month. If a teacher completes a topic, then in the upcoming week she assesses it. For CWD, we made a bit shorter assessment paper that included more MCQs and very few detailed response questions. The teacher discusses it with me and then prepares a paper with mutual coordination (Zareen).*

In this research finding, the assessment is modified and learning expectations are minimized. If the performance is not satisfactory, the same topic is assessed differently in the second test. But they introduce a new topic and send monthly reports to parents.

She further explained regarding the summative assessment:

*In final terms, we assess those topics that are important and those a child needs to know before moving on to the next class. Their [CWSNs] exams are not conducted on a single day. However, we divided them into two parts in two days. These children can't take the exam in a single day. It comprises both verbal and written. The verbal includes spelling, dictation, and poems. The written part includes MCQs, one-word answers, choosing the best answer, and filling in the blanks. We help them to practice with one-line answers, otherwise, they will face problems at a higher level. By the end of grade V, they can write answers on four to five lines (Zareen).*

Although in the abovementioned findings, the learning expectations are compromised and reduced, it was effectively justified in the following statement by a teacher. She stated:

*In the case of CWD, there is the terminology 'Individualized Educational Plan.' IEP stands for 'individualized education program.' An IEP is a printed document for a CWD that is developed, revised, and reviewed with consultations in a meeting following certain requirements of law and regulations. Each child's IEP must contain specific information, as listed within the Individuals with Disabilities Education Act [IDEA], and the most important of all is special education and related services to be provided to the child, including supplementary aids and services (such as a communication device) and modifications to the program or supports for school personnel. It includes modifications to assessment as well (Isra).*

The above excerpt highlights the justification and need for the differentiated mode of assessment, as it can play a vital role in the academic success of CWSN.

Another important suggestion that teachers put forward in the context of the current research study is that there is no certificated recognition for children with disabilities if they have not completed matriculation, even if they have reached that level. There must be some sort of incentive in the form of certificates for CWSN if they manage to attain a

certain level of success despite not passing matriculation exams. This will boost them and encourage them.

It is an opportunity that the school is very well aware of the concept of modified assessment and the school has a provision for it. This can play a pivotal role in the academic success of a child with a disability and, above all, if the school follows international standards in this regard. As stated in IDEA (1997), assessments can also be modified keeping their special needs in view [50].

*5.4. Research Findings—Threats*

The lack of acceptability by society is the key threat highlighted by the director's administration and teachers. He was very much concerned about this, and he believed that this setup was sustainable, and one of the reasons for this is that they have created an environment in which there is acceptance of children with special educational needs. They have sensitized parents, teachers, and students about children with special educational needs and that they are a part of them. A director of administration stated:

> *No one is ready to accept them, not even their parents. The parents say these kinds of statements. Why should we invest in them? This is a dilemma. One of the reasons behind this is that if something happens to the parents of CWSN, how long will they be with them? Who will take care of them? The school caters to them with a mission to offer affordable quality special needs education to disadvantaged learners affected by disabilities. Create a safe and compassionate environment, which encourages pupils to reach their full potential in an atmosphere of mutual respect. Attempt to ensure that students become functioning and independent participants in our society by considering and working on their mental, physical, and social well-being. Empower parents with the knowledge to recognize and make better decisions for their children's well-being and future (Mahmud).*

The above statement resonates with the idea of acceptance and also that children with disabilities become functioning and can contribute to the well-being of society. The Praxis School is working for this cause to empower them, as they consider them as human resources [14]; however, the role of parents is very important to be understood, as sometimes parents are not ready to invest in them, meaning they are not accepting of them. The Praxis School counsels them to understand the reality, that if these children are not cared for effectively, who will take care of them in the future? Therefore, it is necessary to accept them and invest in them by considering them as human resources. A teacher stated, "*They need support and encouragement to be able to emerge as useful members of society*" (Aiman).

Children with special educational needs not only bear the problems of their disability but frequently face negative attitudes from society. Stigmatization involves labeling, bias, and, ultimately, exclusion. Corrigan (2004) has defined two types of stigma: public stigma and self-stigma [51]. Although public stigma involves negative perceptions by society of an individual, considering them socially unacceptable, self-stigma is the individual's self-labeling as socially unacceptable. Both types of stigma cause the unacceptance of children with special educational needs. It is the most terrible threat, which is traumatic for CWSN.

A science teacher stated:

> *The biggest threat is acceptance. If a child with special needs is admitted, then parents of regular children think that they will learn something wrong from them. Now, convincing them is the biggest challenge. These children without disabilities keep listening to these sorts of statements at home. Do not sit with CWSN. Do not talk to them. Unfortunately, there is a stigma attached to them (Yasra).*

The above statement highlights the need for acceptance; if CWSN are not accepted then the school believes that successful inclusion cannot happen. A mathematics teacher mentioned that parents have a "negative mindset" (Isra). She further added:

*Parents can be the biggest barrier if they are not cooperating. Sometimes they feel reluctant to send CWSN to an inclusive school because people will think that their child is not normal. This negative mindset seems an invisible threat (Isra).*

With this sort of mindset, CWSN will not be able to obtain recognition and representation [52]. Fraser (2007) emphasized that to recognize an individual, acceptance and inclusion are essential and lead to representation [52]. When CWSN live in a society with these perceptions, they are often subject to stigma and social exclusion and are not recognized. The social consequences of stigma distress the CWSN and this extends to the family, whose entire social status comes under threat [14] and leads to misrepresentation [52].

The teachers at the Praxis School repeatedly reported that parents of children without disabilities showed their concern. A coordinator of the inclusive setup stated:

*The parents created a fuss. Why did you make my child sit with a special needs child? They complain that our child picked up his/her bad habit. I told them that sometimes a regular child has the worst habit that a child with a disability does not have. The parents need to be counseled (Zareen).*

Negative attitudes held by parents toward children with special educational needs seem to be based on a lack of awareness and may largely be shaped by awareness campaigns. It is well documented that campaigns increase awareness and knowledge regarding children with disabilities and are effective at reducing stigma, thus resulting in increased acceptance [14]. The Praxis School believes that these campaigns are also a source of counseling for parents, as they make them aware of disability and its acceptability.

A coordinator stated, "*These campaigns are organized at the school quite frequently where counselors try to reduce stigmatization and as a consequence, acceptability is increased. The setup sustains because of these campaigns and workshops*" (Rushna).

Therefore, it is desperately required to create awareness among people that CWSN is a part of us. We need to accept them by creating welcoming spaces for them just like the Praxis School.

## 6. Discussion

The emphasis of this section is on the pedagogical practices adopted by teachers at the Praxis School. In the course of the thematic analysis, it was found that active learning strategies seem to be a good idea, as the students become engaged and settle effectively in an inclusive setup. This finding is consistent with the literature in that activity-based learning is a key finding when it comes to accommodating children with special educational needs [14]. As is evident from the findings, for teaching a single concept, they need multiple activities. Children with special needs require lots of repetition and revision. The activities help with retention as they have short-term memory. Another key area is modification. They do not introduce anything new in order to teach them. They modify the syllabus by skipping all unnecessary details and teaching them in a simplified manner. They give short commands to them and teach them in steps. They simplify content as much as they can, and this is modification. However, the universal design of learning is followed at the Praxis School, which is a set of principles that outline the idea that every person has a unique learning style and there should be opportunities open to every student to acquire knowledge in their own manner [46]. For this reason, at the school, the three main components of any lesson (material representation, activities with material, and engagement) are shaped following various students' needs and challenges to create an effective learning environment. This finding is also consistent with the literature in that activities engage and help to reduce disruptive behavior [29]. Alternative teaching is observed at the Praxis School, which presumes that one teacher works with a small group in a class, while the other works with the remaining students. The incorporation of modified assessments is an opportunity for a child whose intellectual, motor, and perceptual skills are hampered. This finding is consistent with the literature in which modified assessments are ensured [14]. However, children with disabilities have to struggle with assistive devices

as the school lacks them. This finding is consistent with the literature related to a context of developing countries; however, it is inconsistent regarding developed nations [10].

In numerous cases, as the data have shown, there is no "one plan fits all" for determining how teachers should cater to the aggressive behavior of students with disabilities in inclusive classrooms. An initial starting point would include establishing classroom rules and regulations, demarcating classroom limits, setting expectations, stating responsibilities, and developing a meaningful and functional curriculum in which all children can receive teaching–learning experiences that can be differentiated, individualized, and integrated. However, it is apparent that several participating coordinators and teachers believed that their school did as much as they could to support children with special educational needs in harmony with their mission statement, and the research findings reveal that, in a broad number of cases, their practice reflects these beliefs. But, on the other hand, the lack of acceptability by society is a major threat to children with disabilities. There is a need to accept them; only then can the agenda of inclusion be transformed into reality worldwide.

## 7. Delimitations and Limitations of the Study

In this study, an embedded single case study design was employed, which had both strengths and limitations. One limitation inherent in the qualitative research, including this particular study, is its incapacity to be generalized to a broader population [38]. Replication proves challenging due to contextual differences, and researchers interpret data from their unique perspectives [45].

Despite these challenges, the research uniquely focuses on special educational needs in Pakistan, presenting recommendations to address educational gaps. The practical implications of the research involve disseminating positive practices to policy makers and schools. Notably, the study identifies a gap in prior research in Pakistan, which has not explored the link between a holistic school approach and teaching practices, especially in inclusive settings for middle- to lower-middle-class students [12].

By examining such practices in a distinctive school environment, this study fills that void. Although limited to a single school, it unveils valuable insight into pedagogical practices within an inclusive private school. While the findings cannot be broadly applied, they bear relevance for similar schools. The study narrows its focus to primary-level teachers and leaders (grades I to V), rendering its conclusions inapplicable to other education levels.

The data collection involved self-reported interviews and classroom observations. The time constraint prevented independent data verification, but cross-referencing of interview and observation data enhanced the results' reliability.

**Author Contributions:** Conceptualization, M.K.; methodology, M.K.; validation, N.B.; formal analysis, M.K.; investigation, M.K.; resources, N.B. and S.S.; writing—original draft preparation, M.K.; writing—review and editing, N.B. and S.S.; visualization, M.K.; supervision, N.B.; project administration, N.B.; funding acquisition, S.S. All authors have read and agreed to the published version of the manuscript.

**Funding:** This research received no external funding.

**Institutional Review Board Statement:** The researchers adhered to fundamental ethical principles and the APA's ethical code. The study involving human participants did not necessitate ethical review and approval in accordance with local legislation and institutional requirements. However, the entire study and questionnaire underwent design and review under the supervision of the research team led by the first and second authors, comprising educationalists from a private university in Metropolis City, Pakistan, well-versed in the country's educational system. The reviewers detected no potential conflicts of interest or harm to participants, and the activities remained within the bounds of ethical conduct. Research participants were informed about the study's purpose, methodology, and the intended use of data. The researchers clarified the participants' role in the research and how it could contribute to academia. It was ensured that confidentiality and anonymity were upheld, and participants were made aware that their involvement was voluntary, with the right to withdraw at any time. Participants were reassured that risks or harm would be avoided in all instances. Additionally, they were briefed on secure data storage during and after the project. Access to the data was restricted

to researchers and supervisors only, with measures in place to ensure privacy and confidentiality. The use of pseudonyms was explained to schools, teachers, head teachers, and students to prevent identification during the reporting of research findings.

**Informed Consent Statement:** Informed consent was obtained from all participants involved in the study.

**Data Availability Statement:** All relevant data are cited and referred to in the text.

**Acknowledgments:** The researchers acknowledge the support from the German Research Foundation and the Open Access Publication Fund of TU Berlin.

**Conflicts of Interest:** The authors declare no conflicts of interest.

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
