# Peer review of "A SWOT: Thematic Analysis of Pedagogical Practices at Inclusive School of Pakistan"

_societies, doi:10.3390/soc14020021_

Round 1

Reviewer 1 Report

Comments and Suggestions for Authors

Dear authors,

The article captivates with its meticulous approach to outlining the research method and aim. The high priority given to the theme, coupled with the multi-layered analysis, adds significant depth to the narrative.

However, it is worth noting that a potential enhancement to the article could involve a more comprehensive exploration of existing literature on inclusive pedagogy. Drawing on the works of scholars like Lani Florian, Tracy Edwards, and their colleagues would not only enrich the discussion but also provide a more nuanced context for the research.

Moreover, the suggestion to shift the focus from an old paradigm, placing ownership of the problem on the child, to a more holistic examination of the context is insightful. The article could benefit from a broader perspective that emphasises the idea that what works for children with disabilities is beneficial for all children. Inclusive pedagogy and didactics are not just advantageous for children with specific needs; they bring enrichment that is valuable for all students.

While the authors touch upon the concept of acceptance, there is a compelling argument for expanding the narrative to emphasise a broader goal. The objective should extend beyond mere acceptance; instead, it should center on recognising the necessity for all children to contribute to building the society of tomorrow. Every child, irrespective of their abilities or backgrounds, brings a unique richness that is inherently valuable.

Finally, efforts should be directed towards improving the visual representation of data. Taking steps to streamline these visual elements will undoubtedly elevate the scholarly impact of the article.

In conclusion, the article is a commendable exploration into the world of inclusive education. Strengthening the link to existing literature and broadening the perspective to highlight the universal benefits of inclusive practices could further elevate the impact and relevance of this noteworthy research work.

Author Response

All the comments are addressed in the attached file.

Reviewer 2 Report

Comments and Suggestions for Authors

Dear colleagues, thank you for this much needed study in the given context. Hereby, I would like to point to some issues that may be improved. 

1. Inclusive education is presented as "putting children with and without disabilities in the same physical environment" throughout the manuscript. This usage is diverging from the concept of inclusion "as providing the best suitable response to the needs". Achieving physical inclusion is only one dimension among many. Unfortunately, the way inclusion / inclusive education is presented in the manuscript is rather limited and not comprehensive. 

2. The reason behind conducing a SWOT analysis is not discussed thoroughly. Are there similar studies using this method? How this method can contribute to the knowledge in an innovative way? Supporting the usage of this method with literature is recommended. 

3. Ecological System Theory is explained very clearly. However, the connection of this theory and this study remains vague. How are the components of this theory used in this study? How is this theory applied during this study? Right now, it is a bit difficult to understand the justification of explaining this theory in the literature review part. 

4. In the literature review part, there is an emphasis on mental health. However, the study does not deal with mental health only. This excessive emphasis on mental health and the research context (data) do not relate to each other fully. 

5. Table 2 should be reformatted. The table is not reader friendly. 

6. Figure 5 seems to be incomplete. Especially the red parts. 

Comments on the Quality of English Language

Please run a test especially for the usage of definite and indefinite article. 

Author Response

(The authors gave the same response as above.)

Round 2

Reviewer 1 Report

Comments and Suggestions for Authors

Dear Authors

I hope this message finds you well. I am delighted to inform you that after carefully reviewing your manuscript titled "A SWOT: Thematic Analysis of Pedagogical Practices at Inclusive School of Pakistan," I find that it has been significantly improved and now better situates itself within the field and existing literature, making it suitable for publication in Societies.

Your revisions have substantially strengthened the manuscript's coherence and relevance to the broader academic discourse. I commend your efforts in addressing the reviewers' comments and integrating additional insights from the literature. I am confident that its contributions will enrich the scholarly dialogue within Societies and beyond.

Thank you for the opportunity to review your work. I eagerly await the next steps in the publication process.

Warm regards

Author Response

Thank you for the appreciating comments.

Reviewer 2 Report

Comments and Suggestions for Authors

Dear colleagues,

thank you for the revisions. The manuscript is definitely improved. Here are my suggestions for minor revisions

1. Listing challenges associated with inclusive education steps in the section 2.2 can be reconsidered. Instead of a list where each challenge relies on a single source, having a paragraph where these challenges are discussed would make the flow more smooth.  

2. The presentation of the result can be adapted. It is not easy to follow the flow of the results. Each passage is followed by a quotation, however the relevance between the passages and the quotations is not all the time visible. Adding sentences that point to the relevance more explicitly is suggested. 

Comments on the Quality of English Language

None

Author Response

Issues are addressed and explained in the attached file.
